# Neurodevelopmental Outcomes in Tetralogy of Fallot: A Systematic Review

**DOI:** 10.3390/children9020264

**Published:** 2022-02-15

**Authors:** Kalliopi Kordopati-Zilou, Theodoros Sergentanis, Panagiota Pervanidou, Danai Sofianou-Petraki, Konstantinos Panoulis, Nikolaos Vlahos, Makarios Eleftheriades

**Affiliations:** 12nd Department of Obstetrics and Gynecology, Aretaieio Hospital, National and Kapodistrian University of Athens, 11528 Athens, Greece; kpanoulis@med.uoa.gr (K.P.); nikosvlahos@med.uoa.gr (N.V.); melefth@med.uoa.gr (M.E.); 2Department of Hygiene, Epidemiology and Medical Statistics, School of Medicine, National and Kapodistrian University of Athens, 11527 Athens, Greece; tsergent@med.uoa.gr; 3Unit of Developmental and Behavioral Pediatrics, First Department of Pediatrics, “Aghia Sophia” Children’s Hospital, School of Medicine, National and Kapodistrian University of Athens, 11527 Athens, Greece; ppervanid@med.uoa.gr; 4First Department of Pediatrics, “Aghia Sophia” Children’s Hospital, School of Medicine, National and Kapodistrian University of Athens, 11527 Athens, Greece; danaisofianoupetraki@gmail.com

**Keywords:** tetralogy of Fallot, neurodevelopment, neurodevelopmental disorders

## Abstract

BACKGROUND: Tetralogy of Fallot (TOF) represents between 7 and 10% of the total cases of congenital heart defects (CHD) and is estimated to be the most common cyanotic CHD, requiring medical or surgical intervention within the first year of life. Current advances in prenatal screening and fetal echocardiography led to increased rates of prenatal diagnosis of TOF. Furthermore, improvements in initial medical care, surgical repair, and long-term care are associated with excellent long-term survival until adulthood. Consequently, issues of morbidity have come under the spotlight, specifically neurodevelopmental and psychiatric adverse outcomes, which affect the quality of life of TOF survivors. METHOD: This study is a systematic review of English articles, using PUBMED and applying the following search terms, Tetralogy of Fallot, neurodevelopment, autism, cerebral palsy, attention-deficit hyperactivity disorder. Data were extracted by two authors. RESULTS: Most researchers suggest that TOF survivors score lower in neurodevelopmental tests than healthy populations of the same age and are in danger of neurodevelopmental impairments. Furthermore, it is suggested that TOF adolescents show higher rates of psychiatric disorders. CONCLUSIONS: The neurodevelopment of TOF survivors is not intensively studied. Existing studies in TOF survivors focus on different developmental aspects, using different evaluation methods and thus making conclusions for either one of the four aspects of neurodevelopment (executive function, cognition, and adaptive function, speech-language and motor function, or neuropsychiatric domain). The poor outcomes of these isolated studies indicate the need for future research as well as for continuous neuropsychological assessment and close monitoring of children and adolescents with TOF.

## 1. Introduction

Tetralogy of Fallot (TOF) appears to be the most frequent cyanotic congenital heart defect (CHD), as it accounts for 7–10% of all CHD. Its prevalence is calculated at about 4 per 10,000 live births [1]. Within the present surgical era, infants with TOF undergo complete intracardiac repair early in their lives. Long-term survival rates are high, approximately >90% after 25 years of surgery [2]. Although the long-term morbidity has been almost eliminated after surgical TOF repair, there is growing consent that these patients are in danger of neurodevelopmental (ND) impairment.

Preoperative and intraoperative features have been correlated to adverse ND outcomes in the TOF population. Preoperative factors, such as chronic hypoxemia, hypoperfusion, acidosis, and thromboembolic events, have been related to ND impairments in this patient group [3,4]. Furthermore, the duration of cardiopulmonary bypass (CPB), use and duration of deep hypothermic circulatory arrest (DHCA), hemodilution, degree of hypothermia, blood gas management, and other intraoperative management strategies have been implicated as potential contributors to cerebral injury [5,6,7,8]. However, as the aforementioned risk factors do not fully interpret the differences in ND outcomes observed among TOF patients, the pathogenesis of adverse ND sequelae is considered multifactorial.

TOF is observed as a part of many genetic syndromes and chromosomal anomalies related to impaired neurodevelopment, either as a phenotypic feature or as an associated clinical finding [9]. Specifically, 16% of patients with TOF are diagnosed with 22q11 microdeletions, trisomy 21, Alagille’s syndrome, Cat Eye syndrome, or CHARGE and VATER associations [9,10,11,12,13]. Additionally, genetic polymorphisms exist that could impair neuro resiliency and increase susceptibility to neurologic injury after infant heart operation [14,15]. 

In the existing literature, there is a limited number of studies concerning TOF survivors’ neurodevelopmental performance solely. Moreover, each study uses a unique validated scale to assess ND, mainly in line with the region that is being assessed and the ND domain examined. In addition, each research focuses on a specific age population. Consequently, this systematic review aims to gather and summarize all the prevailing data concerning the neurodevelopmental course of TOF patients. So as to attain that, we addressed the subsequent questions: (1) Are TOF survivors’ neurodevelopment affected at every age? (2) Which validated scales are used to assess neurodevelopment? (3) Is there a difference within the performance demonstrated between TOF survivors and a group of healthy peers or mean population? (4) Which particular domains are mostly affected and which are not affected?

## 2. Materials and Methods

### 2.1. Search Strategy

This review was performed in step with a protocol invented a priori. PUBMED was electronically searched with the use of combinations of the applicable keywords and word variants for ‘tetralogy of Fallot’, ‘neurodevelopment’, ‘autism’, ‘cerebral palsy’, ‘attention deficit hyperactivity disorder’. The language that was imposed on the choice criteria were English. Additional reports were hand sought in reference lists of relevant articles and reviews.

### 2.2. Study Selection

Studies included during this systematic review were consistent with the following criteria: population, outcome, and study design. Eligibility criteria for the population of enrolled studies were: patients with tetralogy of Fallot after cardiac surgery, with or without pulmonary atresia, and normal chromosomal status (46, XY, and XX). Patients with microdeletion 22q11.2, or other non-numerical chromosome abnormalities, were included. TOF appeared sporadically in the families studied. At the time of evaluation, TOF survivors were in good overall health condition, including their cardiac status. A number of them received cardiac-related medication. The repair of patients’ cardiac defects is considered to be anatomically and functionally adequate. Neurodevelopment of the population mentioned above was evaluated by a great variety of validated scales, according to the study region, the age of TOF survivors at the time of the study, and the neurodevelopmental domain assessed. All different scales were enrolled during this review. As far as the outcomes derived from the selected studies, values were presented as mean SD for continuous variables. Data were addressed as percentages or numbers of affected children for categoric variables and outcomes. Case-control studies, prospective and retrospective cohort, and case series were included.

### 2.3. Data Extraction

All records were independently reviewed by two authors. Initially, they searched the database and retrieved full texts of the studies that met the inclusion criteria established according to the PICOS approach. Concordance about potential relevance was met by consensus. Afterwards, two authors independently reviewed the papers and summarized the findings on data extraction forms with respect to study characteristics, outcomes, and quality. Reviewers discussed inconsistencies and a consensus was reached.

## 3. Results

The database search yielded 101 possible papers. Of these, 77 were discarded by irrelevant titles or abstracts. Three duplicates were found. Full manuscripts were retrieved for 21 studies. It appeared that 14 met the eligibility criteria as described above and were included in the systematic review. The 7 excluded studies did not evaluate neurodevelopment of TOF patients compared to the mean population but other CHDs or used imaging tools or other physical examination tests to present results. No unpublished relevant studies were obtained. (Figure A1) (Table A1). Main patient, operation and medical history characteristics of TOF patients within the studies are presented in Table A2.

Of the studies assessed for this review, two focused on infants surviving TOF [16,17]. Zeltser et al. during a cohort study, evaluated 60 infants at 12 months old, using the Bayley Scales of Infant Development-II, which consists of two separate scores, the Psychomotor Developmental Index (PDI) and the Mental Developmental Index (MDI) [17]. The study concluded that most of TOF survivors had neurodevelopmental scores within the normal range 1 year after their surgical repair, highlighting that the presence of a genetic syndrome constitutes a risk factor for lower MDI and PDI scores. (*p* = 0.002 and *p* = 0.001)

Favilla et al. examined 49 children with repaired TOF via the Bayley Infant Neurodevelopmental Screener (BINS), Peabody Developmental Motor Scale (PDMS), and Bayley Scales of Infant and Toddler Development Third Edition (Bayley-III) [16]. In the study, 43% of patients (*n* = 16) had shortfalls in the BINS, the primal screening test, and in particular gross motor deficits, deficits in receptive language, cognitive and fine motor skills. As far as the PDMS and the Bayley-III are concerned, the median scores were found to be lower than that of the peers.

While assessing children during their preschool years (4 years old), Gaynor et al. followed-up 44 children, excluding those with congenital anomalies [18]. With the use of a wide range of scales, which examined most of the domains of neurodevelopment, Gaynor et al. concluded that the mean unadjusted results derived from the assessment of all ND domains were within the overall population’s anticipated range (±SD). However, they noted that moderate to severe impairment in a minimum of 1 ND domain was found to be of a higher rate within TOF survivors than expected for the overall population.

Hovels-Gurich et al. studied 20 TOF survivors with hypoxemia and no syndromic form, operated at a mean age of 0.7 (SD 0.3) years with the use of deep hypothermic circulatory arrest and low flow cardiopulmonary bypass [19,20,21,22]. They examined the participants at mean age 7.4 (SD 1.6) years by scales that evaluate all the different aspects of neurodevelopmental progress. At first, TOF survivors underwent standardized evaluation of their neurologic status, intelligence, academic achievement, language, gross motor function, and exercise capacity [21]. Socioeconomic status and exercise capacity did not differ from healthy children. Whereas motor function, formal intelligence, academic achievement, expressive and receptive language were found to be statically significant lower (*p* < 0.01 to *p* < 0.001). Subsequently, Hovels-Gurich et al. published a study that aimed to objectively estimate the long-term functions of attention of TOF children by means of the Attention Network Test (ANT), a neuropsychological inventory assessing three anatomically and functionally defined attentional networks [20]. Conflict performance was estimated to be significantly reduced within the TOF group (*p* = 0.005), contrary to alerting and orienting, which were found to be within the conventional range. In the same year, a different study introduced the impact of CHD in TOF survivors’ behavioral and emotional development as well as the quality of life (QoL) [22]. Compared to healthy children, internalizing and externalizing issues were higher among the TOF group, accomplishments at school and overall competence were lower, counter to self- and parent-reported quality of life that was found to be adequate.

Last but not least, Hövels-Gürich et al. assessed the speech and language development of TOF survivors at a mean age of 7.4 years [19]. Total scores of children with TOF on oral and speech apraxia (Mayo Test), as well as on oral and speech motor control functions (TFS), were significantly impaired (*p* < 0.02 to < 0.05) compared to their peers. Phonological awareness and auditory word recognition were described as normal when evaluated by the Test of auditory analysis skills and the Auditory Closure subtest of the Illinois Test of Psycholinguistic Abilities, respectively.

Taking into consideration the group of school-aged children, Miatton et al. examined 18 patients of 8 years old making use of Wechsler Intelligence Scale for Children (3rd edition, Dutch version), a neuropsychological assessment battery (NEPSY) and the Child Behavior Checklist (CBCL) [23]. The application of the assessment tools revealed that children with TOF faced reduced full-scale intelligence than the healthy control group (*p* < 0.05) and a neuropsychological profile defined by mainly mild motor impairments (*p* < 0.01) and frustration with language tasks (*p* < 0.01). In addition, parental reports declared significantly elevated scores on attention problems (*p* < 0.05), and total problem scale (*p* < 0.05), and classified the child’s school competencies as reduced compared to healthy peers (*p* < 0.01).

Six different studies concerning the age group of adolescents were retrieved [24,25,26,27,28,29]. Bellinger et al. in a case report study, enrolled 91 TOF patients (13–16 years old) and 87 referent subjects [24]. They assessed neurodevelopment via tests concerning executive functions, visual-spatial skills, memory, attention, academic achievement and social cognition. In all of the aforementioned domains, patients scored lower than their healthy peers, regardless of their genetic status. However, in most respects, patients that carried out significantly lower scores were those with a genetic/phenotypic diagnosis. 

Bean Jaworski et al. included in their research the group of TOF patients from Bellinger’s study and aimed to assess the deficits in visuospatial processing and academic achievements [28]. They indicated greater odds of impairment in all trials of Structural Accuracy and in the Copy trial of Incidental Accuracy (*p* ≤ 0.5).

Cassidy et al. focused on the impact on specific aspects of neurodevelopment [25,26]. Specifically, they enrolled 68 TOF survivors at 13–16 years of age and conducted 2 different studies. In 2015, they investigated the risk of Executive Function (EF) deficits by using the D-KEFS for laboratory EF tasks and the BRIEF questionnaire to seek facts about the use of EF skills in real-world conditions [26]. Impairments were reported in flexibility/problem-solving and verbally mediated EF abilities as well as in visuo-spatially mediated EF skills. In 2017, Cassidy et al. looked into memory processes and concluded that TOF adolescents scored below the expected population mean on most Children Memory Scale variables, except for the Sequences and Stories: Immediate subtests [25]. Visual-spatial memory deficits both for immediate and delayed memory, were also declared. 

Holland et al. attempted to explore psychiatric disorders and function in adolescents with surgically repaired TOF, including those diagnosed with a genetic syndrome [27]. They revealed that the lifetime prevalence of anxiety disorder was higher (43%) and global psychosocial functioning was reduced in TOF survivors with a genetic diagnosis compared to TOF patients without genetic diagnosis (*p* = 0.04 and <0.001, respectively) and compared to referents (*p* = 0.001 and <0.001, respectively). However, both syndromic and non-syndromic TOF adolescents faced worse rates on parent-/self-reports for anxiety and disruptive behavior. Most significantly, they presented with an increased lifetime prevalence of ADHD (Attention Deficit- Hyperactivity Disorder) in comparison to their healthy peers (*p* = 0.04 and 0.002, respectively). The same outcome was also illustrated from Holst et al. when investigating the prevalence of ADHD symptoms in non-syndromic TOF patients [29]. Inattention symptoms were almost six times more frequent, whereas hyperactivity/impulsivity symptoms were infrequent. Subsequently, a significantly higher total mean score of ADHD occurred for TOF patients than for the control group (*p* = 0.004).

## 4. Discussion

The information extracted from this systematic review indicates that there is an inconsistency concerning the neurodevelopmental assessment of TOF survivors. In particular, from the 14 different studies retrieved occurs that ND was assessed using identical or similar tools only within the infant group. Concordance was not found within the population’s selection criteria, as some studies included and others excluded patients with syndromic/genetic features. Nevertheless, although the majority of children and adolescents with TOF had neurodevelopmental scores within the mean range, there was a substantial proportion who experienced difficulties in meeting neurodevelopmental expectations and manifested deficits compared to the normative population. 

BINS and PDMS executed early in infancy were associated with Bayley-III scores executed after one year of age, confirming that early screening recognizes TOF survivors in danger of impaired neurodevelopment [16]. That being the case, Favilla et al. suggested the application of BINS and PDMS for the discovery of initial deficits, allowing for prompt implementation of rigorous surveillance and therapies, in the same way as what has already been established as standardized care for other CHD patients [16]. When assessing risk factors in infancy that are associated with neurodevelopment, Zeltser et al. revealed that syndromic TOF was a predictor of reduced MDI and PDI (P 0.002 and P 0.001) [17], whereas Favilla focused on the significant impact of socioeconomic parameters and measures of TOF complexity [16]. 

To our knowledge, only Gaynor et al. assessed pre-school TOF survivors among other children with CHD (VSD, TGA, HLHS) after surgery and excluded those with recognized genetic syndromes (18). All of the children included in Gaynor’s study achieved mean scores within the average range for all the neurodevelopmental domains tested. No significant differences emerged when comparing the CHD groups. That was most probably a consequence of the elimination of syndromic patients, therefore underlining, once again, the negative impact of genetic factors on neurοdevelopment.

In a series of studies Hovels-Gurich et al. suggested that school-age children, after cardiac surgery for a congenital defect (TOF or VSD) with the use of cardiopulmonary bypass (CPB) with or without hypothermic cardiac arrest, are broadly in danger of neurodevelopmental deficits in all of the following domains: academic achievement, intelligence, attention, speech, language, motor functions and behavior [19,20,21,22]. Suggested risk factors of neurodevelopmental dysfunction were age at testing, socioeconomic status, and duration of CPB. These studies consistently demonstrated that speech, motor, cognitive and academic disabilities along with executive attentional impairments are significantly correlated. This points out that speech and language restrictions, mainly in the domain of expressive rather than in the field of receptive speech items, can be addressed as a sensitive pointer for global child development after cardiac surgery.

In accordance with previous studies, Bellinger et al. highlighted that a genetic/phenotypic syndrome constitutes a principal determinant of neurodevelopmental impairments [24]. By exploiting MRI, in order to shed light on the anatomy of the patients’ brain, the study revealed that the prevalence of brain abnormalities was not statically different among TOF survivors with and without a genetic diagnosis. Based on that fact, the authors assumed that genetic factors, still unknown, participate in the etiology of TOF and probably affect brain development. These factors might also contribute to the ND deficits observed even in patients without a syndromic form.

Holland et al., along with Holst et al., investigated the psychiatric sequelae of TOF adolescents [27,29]. Although only the first study included children with associated genetic diagnoses, both concluded that the prevalence of ADHD was higher among the TOF population. The precise reasons for the increased psychiatric disorders are suggested to be multifactorial. Patient-specific characteristics, such as genetic factors, delayed fetal brain maturation, impaired executive functioning, increased parental/familial stress and parental over-protection are potential interrelated risk factors. 

Holst et al. also declared that the clinical range of ADHD symptoms is associated with lower quality of life for TOF patients [29]. This highlights the significance of prompt identification and treatment of ADHD, in order to achieve an adequate quality of life for both children and families.

While evaluating and comparing different CHD subgroups during adolescence, Cassidy et al. pointed out that CHD constitutes a predictor for adverse EF development [26]. Memory impairment was mainly associated to socioeconomic status, sex, CHD subgroup and seizure history. Specifically, learning and memory tasks were found to be 2 to 3 times more affected in patients with seizure history. Concerning TOF patients, immediate and delayed verbal memory remained mainly intact, whereas immediate and delayed visual-spatial memory was mainly impaired [25].

It is essential to mention that patients with genetic/phenotypic syndrome were not excluded in 4 of 14 studies [16,17,24,27]. All of these studies agreed to the fact that syndromic patients scored lower than patients without syndrome and were in greater danger of adverse neurodevelopmental outcomes, thus declaring the presence of a genetic syndrome as a risk factor. In 3 out of 4 studies, non-syndromic TOF patients also scored lower than healthy peers [16,24,27]. Only, Zeltser et al. observed normal neurodevelopmental status of TOF patients without genetic syndrome [17]. However, it must be pinpointed that they evaluated patients at the first year of age.

Another fact worth mentioning concerns the studies that included TOF patients with the most extreme form of pulmonary valve atresia (TOF/PA). These patients have lower survival rates and higher percentages of reintervention/reoperation; specifically a 14-year survival rate was estimated at approximately 75% [30,31]. Within this review, 6 of 14 studies included TOF patients with pulmonary atresia (Table A2). Three of the aforementioned studies focused on the effect of TOF/PA on the neurodevelopment. Specifically, both Belinger et al. and Zeltser et al. suggested that the small number of TOF/PA patients did not allow them to make safe assumptions, even though these patients were found to score lower in the ND tests [17,24]. On the other hand, Holland et al. suggested that TOF/PA was not a statistically significant risk factor of anxiety disorder or ADHD [27]. It becomes clear once more that there is a gap in our knowledge concerning the ND of the heterogenous TOF group.

As mentioned above, imaging tools, brain MRI particularly, constitute another evaluation method of neurodevelopment. Two of 14 studies performed brain MRI on the evaluated population [24,27]. They both concluded that patients with and without a genetic/phenotypic syndrome did not statistically differ in the frequency of MRI abnormalities. Mainly, abnormalities were focal or multifocal, with the majority including brain mineralization or iron deposits. Belinger et al. mentioned that the frequency of any MRI abnormality was fivefold greater among the patients than the referents (42% versus 8%, *p* < 0.001), but no statistical significance was met concerning developmental abnormalities in particular [24].

The majority of included studies identified specific neuropsychological shortcomings in children with TOF and thus proposed the application of tailored interventional programs. Many authors noted that long-term neuropsychological sequelae along with school problems might be prevented by means of these programs. 

The limitations of our study should also be noted. Most of them derive from the potential selection bias. The reviewed studies show heterogeneity in the age of follow-up, cohort characteristics, features of TOF patients, assessment tools, and follow-up rate. Moreover, heterogeneity is present in the types of studies themselves, including cohort, case-control, and observational studies. We should emphasize the restricted strength of the last ones mentioned. Furthermore, a confounding factor is that most patients’ surgeries occurred at least a decade ago. Thus, current cohorts of TOF patients might benefit from cutting-edge surgical and medical interventions.

## 5. Conclusions

In conclusion, the neurodevelopment of the majority of TOF survivors has been evaluated as adequate. However, there is a small but sizeable amount that has presented ND impairments, even among patients without a genetic syndrome. Our recommendation along with AHA/AAP guidelines [13], is the application of enhanced neurodevelopmental surveillance for children with TOF and the creation of tailored to each patient intervention, aiming to help them reach their fullest potential. Furthermore, our results suggest that extended research is required for the identification of the exact risk factors and patterns of neurodevelopmental impairments in TOF patients and, therefore, the establishment of prevention and early intervention protocol.

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
