# Peer review of "Neurodevelopmental Outcomes in Tetralogy of Fallot: A Systematic Review"

_children, 2022, doi:10.3390/children9020264_

Round 1
Reviewer 1 Report
Well written mention was given to the continuum of disease spectrum of TET vs TET/MAPCS/PA. Conclusions based on assumption of worsened outcome with genetic disorder but interesting that a significant portion without genetic disorder suffer from similar delays.
Reviewer 2 Report
Interesitng review, but umbulance.
You did not subdived pts with genetic, non chromosomal syndrome, this group is the most at risk of developing neurocognitive defect and psychiatric disorders.
The your reporting causes of neurocognitive delay apply to all pts operated for congenital complex heart defect, non only for Fallot.THEREFORE for these reasons I CONSIDER THE WORK TO BE REVIEWED with great attention.
Reviewer 3 Report
In this review, Kalliopi Kordopati-Zilou et al. described neurodevelopmental outcomes of repaired TOF patients derived from 14 heterogeneous studies (including cohort, case-control and observational studies) with different population’s selection criteria and cohort characteristics, different evaluation methods of developmental outcomes (executive function, cognition, and adaptive function, speech-language and motor function, or neuropsychiatric domain), different follow- up rates.
Strengths of the study include the detailed description of the neurodevelopmental outcomes, addressing an important knowledge gap in the field of pediatric CHDs, also highlighthing the inconsistencies and the need for future research. Weaknesses include heterogeneity of population associated with the uncomplete description of neonatal, genetical, cardiological and surgical history of the enrolled patients, which need to be addressed.
Major points:
- ‘Eligibility criteria for the population of 84 enrolled studies were: patients with tetralogy of Fallot after cardiac surgery, with or without pulmonary atresia’. The population cohort described in the different studies is very heterogeneous. In particular, the authors included patients with Tetralogy of Fallot and Pulmonary Atresia, the last generally having a different surgical management because they may need a neonatal surgical or interventional palliation before repair. This may represent an important bias selection and should be addressed in detail, such as, possibly, any post-operative neurological complication.
- ‘normal chromosomal status (46, XY, and XX, respectively). Patients with microdeletion 22q11.2 were not excluded’. The authors should report the percentage of syndromic patients for each study in the table A1.
- ‘No familial accumulation was observed with respect to tetralogy of Fallot’. The authors should better clarify this sentence.
- ‘Neurodevelopment of the population mentioned above was evaluated by a great variety of validated scales, according to the study region, the age of TOF survivors at the time of the study, and the neurodevelopmental domain assessed. All different scales were enrolled in this review’. Did any study report also a cerebral imaging (MRI or others)?
Minor points:
- ‘Table A1. Overview of selected studies on neurodevelopmental outcomes for D-TGA patients’. Please, change D-TGA with TOF/AP.
- In Table A1 the reported in the populations characteristics column ‘patients with regular form of TOF’. Could they clarify what ‘regular’ means?
